# Racial Differences in Vaginal Fluid Metabolites and Association with Systemic Inflammation Markers among Ovarian Cancer Patients: A Pilot Study

**DOI:** 10.3390/cancers16071259

**Published:** 2024-03-23

**Authors:** Oyomoare L. Osazuwa-Peters, April Deveaux, Michael J. Muehlbauer, Olga Ilkayeva, James R. Bain, Temitope Keku, Andrew Berchuck, Bin Huang, Kevin Ward, Margaret Gates Kuliszewski, Tomi Akinyemiju

**Affiliations:** 1Department of Population Health Sciences, Duke University School of Medicine, Durham, NC 27701, USA; april.deveaux@duke.edu (A.D.); tomi.akinyemiju@duke.edu (T.A.); 2Duke University School of Medicine, Duke Molecular Physiology Institute, Durham, NC 27701, USA; michael.muehlbauer@duke.edu (M.J.M.); olga.ilkayeva@duke.edu (O.I.); james.bain@duke.edu (J.R.B.); 3Division of Endocrinology, Metabolism, and Nutrition, Department of Medicine, Duke University School of Medicine, Durham, NC 27710, USA; 4Division of Gastroenterology and Hepatology, The University of North Carolina at Chapel Hill, Chapel Hill, NC 27599, USA; temitope_keku@med.unc.edu; 5Duke Division of Gynecologic Oncology, Duke University School of Medicine, Durham, NC 27710, USA; andrew.berchuck@duke.edu; 6Kentucky Cancer Registry, University of Kentucky, Lexington, KY 40506, USA; bhuan0@uky.edu; 7Georgia Cancer Registry, Emory University, Atlanta, GA 30322, USA; kward@emory.edu; 8New York State Cancer Registry, New York State Department of Health, Albany, NY 12204, USA; 9Duke Cancer Institute, Duke University School of Medicine, Durham, NC 27710, USA

**Keywords:** biological pathways, cancer disparities, inflammatory biomarkers, race, social disadvantage, targeted metabolomics

## Abstract

**Simple Summary:**

The vaginal microbiome may play a role in racial disparities in ovarian cancer; there is evidence suggesting that it differs according to race and contributes to inflammation by producing or consuming compounds that can shape how ovarian cancer progresses. Our cross-sectional study aimed to investigate the extent to which chemical compounds, or metabolites, from vaginal fluid, also differ by race, and whether these metabolites are linked to systemic inflammation. Our study of 20 White and 16 Black patients identified 1 out of 99 metabolites, arachidonoylcarnitine (C20:4), as occurring at lower levels in Black and higher levels in White patients with ovarian cancer. More than one-third of vaginal fluid metabolites considered showed correlations with biomarkers of systemic inflammation. Our findings suggest that vaginal fluid metabolites likely differ by race, and may play an important role in inflammatory processes, which may be relevant to racial differences in ovarian cancer outcomes. These pilot findings have the potential to improve our understanding of biological mechanisms underlying racial disparities in ovarian cancer but need verification with larger sample sizes.

**Abstract:**

The vaginal microbiome differs by race and contributes to inflammation by directly producing or consuming metabolites or by indirectly inducing host immune response, but its potential contributions to ovarian cancer (OC) disparities remain unclear. In this exploratory cross-sectional study, we examine whether vaginal fluid metabolites differ by race among patients with OC, if they are associated with systemic inflammation, and if such associations differ by race. Study participants were recruited from the Ovarian Cancer Epidemiology, Healthcare Access, and Disparities Study between March 2021 and September 2022. Our study included 36 study participants with ovarian cancer who provided biospecimens; 20 randomly selected White patients and all 16 eligible Black patients, aged 50–70 years. Acylcarnitines (n = 45 species), sphingomyelins (n = 34), and ceramides (n = 21) were assayed on cervicovaginal fluid, while four cytokines (IL-1β, IL-10, TNF-α, and IL-6) were assayed on saliva. Seven metabolites showed >2-fold differences, two showed significant differences using the Wilcoxon rank-sum test (*p* < 0.05; False Discovery Rate > 0.05), and 30 metabolites had coefficients > ±0.1 in a Penalized Discriminant Analysis that achieved two distinct clusters by race. Arachidonoylcarnitine, the carnitine adduct of arachidonic acid, appeared to be consistently different by race. Thirty-eight vaginal fluid metabolites were significantly correlated with systemic inflammation biomarkers, irrespective of race. These findings suggest that vaginal fluid metabolites may differ by race, are linked with systemic inflammation, and hint at a potential role for mitochondrial dysfunction and sphingolipid metabolism in OC disparities. Larger studies are needed to verify these findings and further establish specific biological mechanisms that may link the vaginal microbiome with OC racial disparities.

## 1. Introduction

Ovarian cancer is the most lethal female cancer [1], with disproportionately worse survival for Black patients despite lower incidence [2,3]. Compared to White women, Black women have 26% increased odds of late-stage diagnosis with ovarian cancer [4], and 30% higher risk-adjusted mortality [2,3]. Between 2005 and 2014, ovarian cancer mortality decline rates have been significantly slower for Black women, with rates declining by −1.1% for Black, compared to −2.4% for White women [5]. Moreover, racial and ethnic disparities in ovarian cancer outcomes persist even in equal-healthcare settings or after adjusting for social determinants of health [6,7,8,9]. Therefore, it is necessary to examine potential biological contributors to these disparities. 

One such biological factor might be the vaginal microbiome, given its ability to modulate inflammation through the production and consumption of metabolites [10,11]. The taxonomic composition of the vaginal microbiome varies by race [12,13]. Black women are more likely to have non-*Lactobacillus* or *Lactobacillus iners*-dominant vaginal microbial communities, whereas *L. crispatus*-dominant vaginal microbiota are more common among White women [12,14]. Non-*Lactobacillus* vaginal communities are known to promote local inflammation and have been implicated in adverse gynecological conditions such as pelvic inflammatory disease, endometriosis, and cervical cancer [15,16,17]. However, even the presence of a *Lactobacillus*-dominant microbiome may not be equally protective for Black and White women, as shown by a report of an eight-fold increased odds of developing cervical intraepithelial neoplasia grade 3 for Black women compared to White women with optimal vaginal microbiome [18]. Yet, there has been no investigation of racial differences in the functional properties of the vaginal microbiome and how this might provide biological insights into racial differences in ovarian cancer outcomes.

Additionally, the potential disparate contribution of the vaginal microbiome to inflammatory pathways relevant to ovarian cancer is unclear given that evidence for ovarian cancer is fragmented and scarce relative to other gynecological cancers [19,20]. However, it appears that for ovarian cancer the vaginal microbiome can create a tumor-permissive microenvironment, either directly, or by inducing host immune responses to contribute to cancer-promoting inflammation and cell death avoidance [10,20,21], which in turn drives tumor initiation and progression through increased oxidative stress, DNA damage, and accumulation of mutations [10,22]. Pro-carcinogenic metabolites could be upregulated, such as inflammation-inducing lipopolysaccharides, growth-factor-like lysophosphatidic acid [10], and prostaglandins [11], while antineoplastic metabolites could be downregulated through consumption as an energy source for cancer cells, such as tryptophan and indoleproprionic acid [10]. Moreover, the vaginal microbiome can also trigger inflammation-inducing signaling pathways involving toll-like receptors that promote a pro-inflammatory response and accelerate tumor growth [20]. 

Thus, there is an important gap in the biology of racial disparities in ovarian cancer given the prominent role of chronic systemic inflammation for ovarian cancer [10,11,23], the potential contribution of the vaginal microbiome to systemic inflammation [10,21], as well as the higher risk of inflammation among Black individuals [24,25,26]. The potential interplay between race, a proxy for social disadvantage [27], and vaginal inflammatory processes associated with unfavorable metabolic profiles may be linked to adverse cancer outcomes [28,29]. 

Prior studies have mainly focused on racial differences in the identity of vaginal microbial flora, using taxonomic composition methods [18,30]. However, a promising alternative approach that can identify relevant biologic pathways [31], not evident from the widely applied taxonomic composition approach, is metabolomics. Metabolomics is a technique that simultaneously measures low-molecular-weight metabolites, end products of cellular processes including amino acids, and lipids [32]. Application of the metabolomics approach can reveal a snapshot of biological processes, and elucidate biological pathways [31]. When applied to cervical cancer research, metabolomics demonstrated drastic alterations of the cervicovaginal metabolome (collection of all metabolites) including lipid perturbations, and a strong correlation between anti-inflammatory nucleotides and *Lactobacillus* dominance [15]. For ovarian cancer, the application of metabolomics to profiling of plasma has demonstrated associations between dysregulation of several metabolites and ovarian cancer incidence [33] and survival [34], including abnormal phospholipid metabolisms, aggressive fatty acid β-oxidation, and aberrant piperidine derivatives metabolism [33,34,35]. No study, to our knowledge, has utilized metabolomics in examining racial disparities in vaginal fluid metabolites among women with ovarian cancer. 

In this study, we take a preliminary step towards bridging this gap by examining whether there are racial differences in vaginal fluid metabolites by characterizing the functional properties of vaginal fluid and correlations with systemic inflammation in Black and White women with ovarian cancer.

## 2. Methods

### 2.1. Study Design

This cross-sectional study used cervicovaginal fluid and saliva samples from Black and White patients with ovarian cancer, aged 50–70 years, recruited into the Ovarian Cancer Epidemiology, Healthcare Access and Disparities (ORCHiD) study between March 2021 and September 2022 [36]. ORCHiD was designed to comprehensively characterize healthcare access among ovarian cancer patients and evaluate its impact on the quality of initial treatment, supportive care, and survival, while also collecting biospecimens. Methodological details for ORCHiD are published elsewhere [36]. In this study, we randomly selected 20 eligible White patients and included all 16 Black patients who were within the selected age range and provided adequate biospecimens to ORCHiD. Each participant gave informed consent to participate in the biospecimen sub-study of ORCHiD. 

### 2.2. Biospecimen Collection and Processing

Biospecimen collection occurred after treatment receipt for all 36 participants and involved self-collection of samples following the manufacturer’s instructions. We opted to use saliva samples for inflammatory assays given its less invasive and convenient collection, and its use as a well-established surrogate for evaluating systemic biological factors. Saliva was collected using the DNA Genotek OGR-500 kit. The vaginal swab was collected using the DNA Genotek OMR-130 kit and the swab was immediately inserted into a securely tightened tube with stabilizing solution. Both samples were enclosed in a biohazard bag and returned to ORCHiD in a self-addressed postage-paid return envelope. Upon receipt, biospecimen samples were cataloged, processed, and stored at the facilities of Duke Molecular Physiology Institute (DMPI). Processing of saliva samples involved heating to 50 °C, splitting into aliquots, and storing at −20 °C; vaginal fluid samples were processed within 30 days of receipt by adding 5 µL of Proteinase K (80 mg/mL), subjecting to a vortex, incubation, splitting into aliquots, and storing at −80 °C. 

### 2.3. Analytical Methods 

For the current study, we focused on measuring targeted metabolites in three compound classes that have been previously related to microbiome taxonomic composition in vaginal fluid [37] and also measured some conventional metabolites. A limitation of these measures is that we are unable to establish the source of the metabolites (i.e., host, bacterial, or nutritional). The main exposure variable was self-reported race, categorized into Black vs. White. We performed targeted metabolomics assays on cervicovaginal fluid at the Metabolomics Core Laboratory at DMPI using tandem mass spectrometry. 

Three compound classes of metabolites involved in lipid metabolism, a major energy source in carcinogenesis, were assayed on cervicovaginal fluid, including acylcarnitines (n = 45), sphingomyelins (n = 34), and ceramides (n = 21). 

#### 2.3.1. Acylcarnitines

Acylcarnitines were analyzed using a stable isotope dilution technique and sample preparation methods described previously [38]. The samples were spiked with a cocktail of heavy-isotope internal standards (Cambridge Isotope Laboratories, Tewksbury, MA, USA; CDN Isotopes, Pointe-Claire, QC, Canada) and deproteinated with methanol. The methanol supernatants were dried and esterified with acidified methanol. Mass spectra for acylcarnitines were obtained using a precursor ion scan of *m*/*z* 99. The data were acquired using a Xevo TQD mass spectrometer equipped with an Acquity^TM^ UPLC system and a data system controlled by the MassLynx 4.1 operating system (Waters, Milford, MA, USA). Additionally, 80% methanol was used as a mobile phase. Ion ratios of analyte to respective internal standard computed from centroided spectra were converted to concentrations using calibrators constructed from authentic aliphatic acylcarnitines (Sigma, St. Louis, MO, USA; Larodan, Solna, Sweden) and dialyzed Fetal Bovine Serum (Sigma, MO, USA). 

#### 2.3.2. Ceramides and Sphingomyelins

Ceramides and sphingomyelins were analyzed as described previously [39,40]. The samples were spiked with ceramide d18:1/17:0 and sphingomyelin d18:1/12:0 (Avanti, Alabaster, AL, USA) as the internal standards and extracted overnight with methanol/chloroform (2:1, *v*/*v*) at 50 °C. The samples were centrifuged (10 min at 3000× *g*) and supernatants were subjected to alkaline hydrolysis to remove glycerophospholipids. After neutralization with glacial acetic acid, ceramides and sphingomyelins were extracted by adding chloroform/water (1:1, *v*/*v*). The samples were vortexed and centrifuged (10 min at 3000× *g*). The lower layer was transferred to a vial, dried down with N_2_, and re-suspended in methanol/chloroform (2:1, *v*/*v*) containing 5 mM ammonium acetate. Ceramides and sphingomyelins were analyzed by flow injection tandem mass spectrometry in the positive mode for precursors of *m*/*z* 264 and 184, respectively, using a Xevo TQS spectrometer (Waters, Milford, MA, USA) and 80% methanol/30 mM ammonium hydroxide as the mobile phase.

#### 2.3.3. Conventional Metabolites

We also assayed cervicovaginal fluid for five conventional metabolites known to be potential energy sources (i.e, glucose), involved in lipid metabolism (i.e., glycerol), provide evidence of altered amino acid metabolism (e.g., β-HB and total ketones), or have anti-microbial properties (i.e., lactate), using a Beckman-Coulter Unicel DxC 600 clinical analyzer (Beckman Coulter Diagnostics, Brea, CA, USA) [41,42]. 

#### 2.3.4. Assays for Biomarkers of Systemic Inflammation

For saliva, we performed inflammatory assays using a human proinflammatory 5-plex sandwich immunoassay following the manufacturer’s instructions (Meso Scale Discovery, Rockville, MD, USA). Four cytokines recognized as biomarkers of systemic inflammation were of interest: IL-1β, IL-10, TNF-α, and IL-6 [43]. 

### 2.4. Statistical Analyses 

#### 2.4.1. Descriptive Summary

Patient descriptive characteristics were summarized, comparing categorical variables by race using Fisher’s exact test and Wilcoxon rank-sum test for continuous variables. Fisher’s exact test effectively handles low expected cell counts of less than 5 typical of small-sized datasets. Similarly, the Wilcoxon rank-sum test was suitable since it is a non-parametric test that makes no assumptions about data distribution, given our study’s small sample size for which assumptions of normality could not be adequately assessed [44,45]. 

#### 2.4.2. Data-Processing Steps

Conventional and targeted metabolites data were normalized to the median, missing and zero values were imputed to 1/5 of the minimum value for the corresponding metabolite, log-transformed, and auto-scaled (i.e., subtract the mean and divide by standard deviation). We also filtered out any metabolite with a constant or single value across samples. 

#### 2.4.3. Racial Differences in Vaginal Fluid Metabolites

We used univariate approaches including fold change (on raw data), Wilcoxon rank-sum tests [45], and volcano plots to compare conventional and targeted metabolite concentrations between Black and White patients. We adjusted original *p*-values from the Wilcoxon rank-sum test to control for multiple tests using false discovery rate (FDR) estimation, which is the expected proportion of false-positive test results [46]. Here, we set FDR < 0.05 as the significance threshold. Further, we used multivariate approaches, including a partial least squares discriminant analysis (PLS-DA) using concentrations for all targeted metabolites. We further performed hierarchical clustering analysis (HCA) first by race, and then by two clinical attributes, histology and stage. For HCA, we used the Euclidean distance and ward clustering algorithm, and only included the top 30% of the most variable metabolites based on interquartile range [37]. Lastly, we explored the separation of clusters by race using a projection in pursuit approach, Penalized Discriminant Analysis (PDA). PDA is able to find reliable projections revealing class separations and selects variables important for separating classes in data with many predictors and few observations [47]. A 2-dimensional optimal projection of targeted metabolites data separated by race was found with a lambda value of 0.1 using functions PDAopt and PPoptViz in package PPtreeViz in R statistical computing software (version 4.2.2).

#### 2.4.4. Correlations with Biomarkers of Systemic Inflammation

Cytokine data were similarly pre-processed by normalization to the median, log transforming, and auto-scaling. We compared cytokine levels by race, and tested for associations between each of four cytokines and all metabolites using Spearman’s rank correlation test. Also, we assessed whether the magnitude of these correlations significantly differed by race using Fisher’s *z*-test. For correlation analyses, we only included patients with complete metabolite and cytokine data (n = 35). 

#### 2.4.5. Software

Data-preprocessing and statistical analyses were mainly done in MetaboAnalyst 5 [48], with Fisher’s *z*-test and heat map visualization done in R version 4.2.2 [49], using packages concor, dplyr, tidyverse, pheatmap, and ggfortify. 

## 3. Results

### 3.1. Descriptive Summary

A total of 16 Black and 20 White ovarian cancer patients were included in this exploratory study. Black patients were on average younger (Mean age, years (SD): 60 (5.74) vs. 63 (6.33) for White patients), with a higher proportion of distant-stage ovarian cancer (43.8% vs. 35% for White patients) (Table 1). We examined the distribution of patient factors by race, including post-menopausal status (Black: 87.5%; White: 100%), receipt of surgery (Black: 100%; White: 100%), and chemotherapy (Black: 87.5%; White: 100%), no receipt of radiation (Black: 100%; White: 100%), no antibiotics use (Black: 68.8%; White: 75%), no suppositories use (Black: 100%; White: 90%), and no douching (Black: 81%; White: 90%; Table 1). There was heterogeneity in histological subtypes across race; among Black patients, unknown histology (43.8%) was most common followed by Type I epithelial (37.5%), while among White patients Type II epithelial was most common (60%), followed by Type I epithelial (25%; Table 1). Data pre-processing resulted in filtering out of the dataset one targeted (C20-OH/C18-DC) and two conventional metabolites (3-Hydroxybutyrate (β-HB) and total ketones) due to constant values across all samples.

### 3.2. Racial Differences

In univariate analyses, seven acylcarnitines showed fold changes with at least a two-fold difference (i.e., >2 or <0.5) between Black and White patients, with two being upregulated, and five being downregulated in Black patients (Table 2). The difference in median concentrations between Black and White groups was statistically significant for two targeted metabolites at *p* < 0.05 but was no longer statistically significant when corrected for multiple testing by FDR (Table 2). When fold change and *p*-values from the Wilcoxon rank-sum test were combined in a volcano plot (Figure 1 left panel), only one targeted metabolite (C20:4) remained significantly different between Black and White patients (Table 2; Figure 1 right panel). None of the conventional metabolites or cytokines considered differed significantly by race in the univariate analysis, and so were not further considered in the multivariate analysis. 

In multivariate analyses, the first two components of the PLS-DA on targeted metabolites accounted for 27.9% and 11.4% of explained variances and resulted in two overlapping groups by race (Appendix A). However, the PLS-DA model did not attain statistical significance based on the permutation test for model validation (*p* = 0.91). Hierarchical clustering analysis based on thirty metabolites resulted in two clusters that were relatively heterogeneous when queried by race (Figure 2), histology (Appendix A), or stage (Appendix A). These thirty metabolites represented the top 30% of all metabolites with the largest variability determined by interquartile range. However, given the high-dimension nature of our dataset relative to the sample size (i.e., 100 predictors and 36 samples), for which traditional methods like hierarchical clustering analysis are known to perform poorly [50], we further utilized an exploratory classification method, the Penalized Discriminant Analysis (PDA) [47]. PDA reduces the number of variables using penalized least squares regression with optimal scoring [47]. The best projection using PDA achieved complete separation between Black and White samples, with 30 metabolites showing coefficients > ±0.1 on the first dimension (Figure 3), including the two metabolites that differed by Wilcoxon rank-sum test (Table 3).

### 3.3. Correlations with Biomarkers of Systemic Inflammation 

Overall, 38 targeted metabolites were statistically significantly associated with at least one of the four biomarkers of systemic inflammation considered at FDR < 0.05, with Spearman’s rank correlation coefficient ranging from −0.62 through 0.76 (Figure 4). Of these targeted metabolites, 2 were ceramides, 13 were acylcarnitines, and 23 were sphingomyelins. Among the systemic inflammation biomarkers considered, IL-1 β was associated with 31, IL-10 with 34, TNF-α with all 38, and IL-6 with 36 different targeted metabolites. C20:4, which differed by race based on raw *p*-value (as described above), showed significant positive correlations with all four inflammation biomarkers. As for conventional metabolites considered, only Glycerol showed significant correlations with inflammation biomarkers including IL-10 (r = 0.41, FDR = 0.04), TNF-α (r = 0.40, FDR = 0.04), and IL-6 (r = 0.42, FDR = 0.03). While none of the correlations between inflammation biomarkers and targeted metabolites differed by race using Fisher’s z (*p* > 0.05; Appendix A), 10–16% of these correlations (IL-1β: 5, IL-10: 5, TNF-α: 4, IL-6: 4) were consistently significant at FDR < 0.05 among Black patients (Appendix A), whereas a much higher proportion (39–90%) of these correlations were consistently significant among White patients (IL-1β: 12, IL-10: 28, TNF-α: 35, IL-6: 25).

## 4. Discussion

While racial and ethnic differences in the taxonomic composition of the vaginal microbiota have been previously described [12,14], our investigation adds to the literature by providing evidence that suggests racial differences in vaginal microbiomes are also demonstrable when measured by fluid metabolites. Moderate differences were observed between Black vs. White vaginal fluid metabolites, including seven acylcarnitines with at least two-fold differences, and two acylcarnitines with significant differences based on raw *p*-values in median concentrations; however, these differences disappeared when FDR was considered. Moreover, two distinct clusters by race were recovered with the PDA method, with substantial contributions from 30 out of 99 metabolites. Arachidonoylcarnitine (C20:4), an acylcarnitine and marker of a diverse class of eicosanoid mediators of responses to inflammation, consistently appeared to be different across most analyses. We also observed significant correlations between thirty-eight vaginal fluid metabolites and inflammation biomarkers. To our knowledge, this is a novel investigation of vaginal fluid metabolites from patients with ovarian cancer and suggests that evidence for racial differences in vaginal fluid metabolites might be found in larger-sized future studies. 

Prior studies have applied the metabolomics approach to gain insights into underlying biological mechanisms for diseases. For example, Srinivasan et al., 2015 used metabolomics approaches to show that cervicovaginal fluid metabolites differed by bacterial vaginosis status, which was characterized by lower concentrations of amino acids, higher levels of amino acid catabolites, polyamines, and 12-HETE, a biomarker of inflammation [37]. Additionally, differences in the cervicovaginal metabolome have been detected among women with cervical cancer, characterized by lipid perturbation and a strong correlation of anti-inflammatory nucleotides with *Lactobacillus* dominance [15]. However, no study has applied this approach of metabolomics on vaginal fluid in a racial disparity context, particularly among women with ovarian cancer, for which inflammation is a prominent risk factor. While there have been no previous metabolomics studies on vaginal fluid, there is evidence for metabolite differences by socioeconomic position. For example, Robinson et al., 2021 applied metabolomics on blood serum and showed that among 30,000 European adults, those in the low socioeconomic position category were markedly characterized by higher levels of glycoprotein acetylation, a marker of inflammation, and lower levels of anti-inflammatory markers such as DHA and omega-3-fatty acids [28]. In this study, using race as a proxy for socioeconomic disadvantage, we found Black vs. White differences in the expression of arachidonoylcarnitine, with median concentrations of the metabolite consistently lower for Black patients. Although this difference disappeared with FDR likely due to reduced power typical of a small sample size, this finding hints at potential biological mechanisms that may be relevant to ovarian cancer in several ways. 

First, arachidonoylcarnitine is in dynamic equilibrium with arachidonic acid, a signal for inflammatory responses and apoptosis induction [51,52]. For colon cancer, it has been shown that cellular levels of arachidonic acid regulate apoptosis, such that overexpression of enzymes *COX-2* and *FACL4* promotes carcinogenesis by lowering arachidonic acid levels [51]. Also, non-steroidal anti-inflammatory drugs induce apoptosis by blocking the metabolic removal of arachidonic acid [51]. The *COX-2* gene also tends to be overexpressed in ovarian cancer, suggesting a plausible connection with arachidonoylcarnitine [53], which differed significantly between Black and White patients with ovarian cancer in our study sample. Such decreased levels of a long-chain acylcarnitine may be interpreted as evidence for its greater utilization in the β-oxidation process in mitochondrial energy metabolism pathways, which may support a role for mitochondrial dysfunction in ovarian cancer racial disparities [54,55].

Second, the substantial correlations between vaginal fluid metabolites and systemic inflammation markers in saliva suggest that vaginal fluid metabolites may be related to both local and chronic inflammatory processes. Elevated genital inflammation has been previously reported among patients with invasive cervical carcinoma relative to patients with low and high-grade cervical dysplasia [17]. While we did not directly measure genital inflammation in our study, the modest correlations found in our study indicate that among patients with ovarian cancer, there might be a link between metabolites in cervicovaginal fluid and systemic inflammation. 

Our findings suggest that the sphingolipid’s metabolic pathway might be relevant to ovarian cancer; ceramides and sphingomyelins account for about two-thirds of the correlations between metabolites and inflammation markers in this study. The sphingolipid metabolic pathway has ceramide at its core, which is generated through de novo pathways, sphingomyelin (SM) decomposition, or sphingosine re-acylation [56]. Ceramides either briefly accumulate or are converted to other metabolites such as sphingosine 1-phosphate (SP1), or reversed into SM [56]. However, ceramide and its metabolites have dynamic opposing effects on cancer biology, a phenomenon termed ‘Sphingolipid rheostat’ [56,57]. Whereas ceramides promote cancer cell apoptosis and inhibit cell proliferation and migration, SP1 promotes cell proliferation and angiogenesis [56,57]. Sphingolipids control the balance in cells between proliferation and apoptotic cell death and tend to have cancer-specific effects [56]. Dynamic balance among components of the sphingolipids pathway can be disrupted in cancer, resulting in altered levels of bioactive sphingolipids [56,57]. 

Consistent with these patterns, we show that the direction of correlations was not homogenous for ceramides and SM, with most showing negative associations and a few showing positive associations. SM metabolism is important in cancer progression because reduced SM degradation results in decreased ceramide production [57]. For ovarian cancer, depletion of sphingomyelin synthase 2, which controls SM content, has been shown to suppress survival, growth, and migration of cell lines via disruption of lipid metabolism and mitochondrial function [58]. 

Overall, while it is difficult to offer specific unequivocal explanations for suggested racial patterns in vaginal fluid metabolites and correlations with systemic markers of inflammation based on our pilot study, the evidence is in favor of the biological relevance of these patterns to racial disparities. Further, our findings justify future larger-scale investigations that overcome this pilot study’s limitations.

A major strength of our study is its novelty, as this is the first study to our knowledge that has investigated racial disparities by examining racial differences in vaginal fluid metabolites. Our study provides valuable pilot data for informing future studies and demonstrates the viability of examining the vaginal microbiome’s functional role in ovarian cancer health disparities. These findings should be interpreted in the context of certain limitations; as is inherent to small studies, we had limited power for robust evaluation of associations. However, we used the PDA method, an exploratory classification method that has been shown to overcome the small sample size large predictor problem, and successfully identified two separate clusters by race based on targeted metabolite concentrations. Another limitation is that we considered only two racial groups and three metabolite compound classes involved in lipid metabolism; our results do not provide a comprehensive evaluation of all racial/ethnic groups or the full vaginal metabolome. Our study cohort, while similar in many respects, was heterogeneous in terms of histology, and was missing some important clinical information (e.g., hormone therapy use). Biospecimens were collected after receipt of treatment for ovarian cancer, including chemotherapy, which is known to alter the vaginal microbiome [21], and the exact treatment date for patients in the study cohort was not available at the time of this study. We did not characterize the taxonomic composition or gene expression of vaginal microbiota in the samples, and so cannot directly relate the metabolites quantified to specific vaginal microbiota. Also, the cross-sectional nature of this study means that vaginal fluid metabolites were characterized at a single time point, whereas properties of vaginal fluid can fluctuate over short periods [21]. Lastly, our study is unable to accurately quantitate absolute metabolite concentrations and our study’s power to detect differences may have further been diminished due to the pre-analytic storage conditions at room temperature, which is known to result in metabolite degradation [59]. Nevertheless, this study’s comparisons of targeted metabolites by racial groups are valid because individual differences between samples have been shown to be much larger than pre-analytic storage-induced differences [60]. 

## 5. Conclusions

In summary, we found evidence that suggests vaginal fluid metabolite concentrations likely differ by race, with Black individuals having a unique pattern of some downregulated metabolites including arachidonoylcarnitine, which has a known indirect role in carcinogenesis. Moreover, significant correlations between systemic inflammation biomarkers and targeted metabolites suggest the involvement of vaginal fluid in sustained inflammation, although no racial differences were observed. Based on these pilot findings, we speculate that racial differences in vaginal microbiome metabolites and correlations with systemic inflammation markers may be biologically relevant to ovarian cancer through pathways involving mitochondrial dysfunction and sphingolipid metabolism. Future studies should evaluate whether these findings are reproducible in a larger sample, and establish what their specific biological implications are. This study provides a template for further investigation of how inflammation from vaginal fluid metabolites may be related to disparities in ovarian cancer.

## Figures and Tables

**Figure 1 cancers-16-01259-f001:**
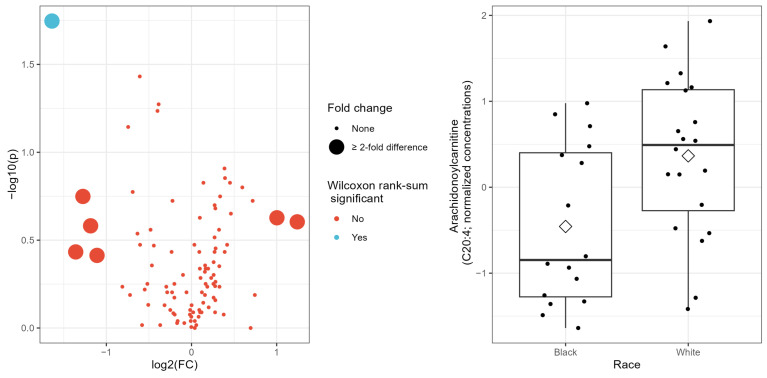
(**Left panel**): Volcano plot showing −log10 *p*-values from Wilcoxon rank-sum test on the *y*-axis and natural log-transformed fold-change on the *x*-axis. (**Right panel**): Boxplot showing normalized concentrations of Arachidonoylcarnitine (C20:4) by race; diamond symbol indicates mean value for race group.

**Figure 2 cancers-16-01259-f002:**
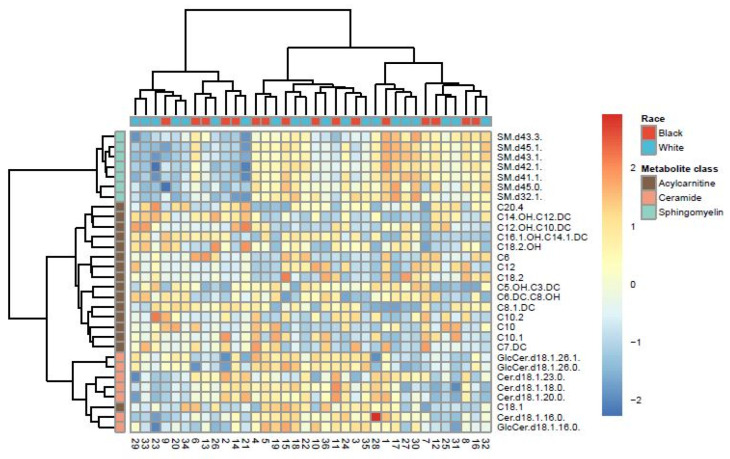
Heat map showing hierarchical clusters by race, based on top 30 targeted metabolites with the largest variability identified by interquartile range.

**Figure 3 cancers-16-01259-f003:**
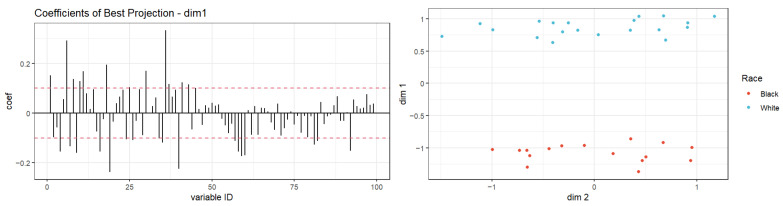
Projection in pursuit index with Penalized Discriminant Analysis using race as a class variable for vaginal fluid metabolite data (*p* = 100, n = 36). Left panel: shows the distribution of coefficients for metabolites in PDA on dimension one of optimal projection, with the dashed red line indicating a threshold of ±0.1. Right panel: shows the separation of clusters by race of projected data onto the optimal 2-dimensional projection using PDA index with lambda = 0.1. Samples are fully separated by race on dimension one.

**Figure 4 cancers-16-01259-f004:**
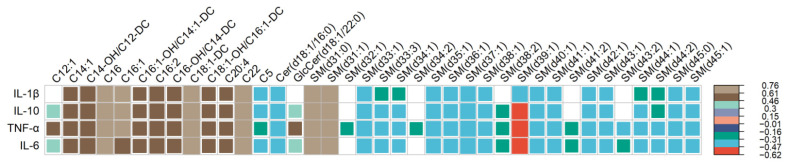
Correlogram showing Spearman’s correlation coefficients (depicted by colored squares) that reached significance threshold at FDR < 0.05, for estimates of the association between each systemic inflammation biomarker (in rows) with targeted metabolites (in columns). The legend on the right side of the plot specifies the corresponding correlation coefficient for each color. White squares indicate no correlation between inflammation biomarkers (row) and metabolite (column).

**Table 1 cancers-16-01259-t001:** Patient characteristics summarized by race.

	Categories	Black	White	*p*-Value **
Sample size		16	20	
Age, years (Median (25th, 75th))		60 (55.75, 63)	64.00 (59.5, 69)	0.122
Histology (%)	Type I epithelial	2 (12.5)	5 (25.0)	0.123
	Type II epithelial	6 (37.5)	12 (60.0)	
	Other	1 (6.2)	0 (0.0)	
	Unknown	7 (43.8)	3 (15.0)	
Stage (%)	Local	0 (0.0)	3 (15.0)	0.542
	Regional	2 (12.5)	2 (10.0)	
	Distant	7 (43.8)	7 (35.0)	
	Unknown	7 (43.8)	8 (40.0)	
Chemotherapy receipt (%)	No	2 (12.5)	0 (0.0)	0.19
	Yes	14 (87.5)	20 (100.0)	
Radiation receipt (%)	No	16 (100.0)	20 (100.0)	
Surgery receipt (%)	Yes	16 (100.0)	20 (100.0)	
Antibiotic use (%)	No	11 (68.8)	15 (75.0)	0.498
	Yes	1 (6.2)	3 (15.0)	
	Unknown	4 (25.0)	2 (10.0)	
Vaginal suppository use (%)	No	16 (100.0)	19 (95.0)	1
	Yes	0 (0.0)	1 (5.0)	
Vaginal douching (%)	No	13 (81.2)	18 (90.0)	0.637
	Unknown	3 (18.8)	2 (10.0)	
Menopausal status (%)	Post-menopausal	14 (87.5)	20 (100.0)	0.19
	Unknown	2 (12.5)	0 (0.0)	

Histological categories represented in the study cohort: Type I epithelial: clear cell carcinoma (ICD-O-3 code: 8310), endometrioid carcinoma (8380), and mucinous carcinoma (8480); Type II epithelial: serous carcinoma (8441, 8461), mixed epithelial-stromal carcinoma (8323, 8980), and undifferentiated or other epithelial (8140); Other: sex cord-stromal (8620). ** *p*-value was obtained from Fisher’s exact test for all categorical variables.

**Table 2 cancers-16-01259-t002:** Important metabolites identified by univariate analyses including fold change, Wilcoxon rank-sum test, or both.

Metabolite	Fold Change		Wilcoxon Rank-Sum Test		
	FC	Log2(FC)	V	*p*-Value	−Log10(P)
C20:4	0.32113	−1.6388	86	0.018	1.747
C4-OH	0.65631	−0.60754	94	0.037	1.431
C14:2	0.38962	−1.3598	131	0.369	0.433
C16:2	0.4131	−1.2754	117	0.178	0.749
C6	2.3646	1.2416	197	0.249	0.604
C22	0.43984	−1.1849	124	0.262	0.582
C18:2-OH	0.46345	−1.1095	132	0.386	0.413
C7-DC	2.0041	1.0029	198	0.236	0.627

Note: Metabolites in highlighted rows showed at least two fold differences (i.e., FC ≥ 2 or ≤ 0.5), or *p*-value < 0.05 for Wilcoxon rank-sum test. For example, C4-OH showed significant difference by Wilcoxon rank-sum test (highlighted in grey), but did not attain absolute FC ≥ 2 (not highlighted). No metabolite reached FDR < 0.05 for Wilcoxon rank-sum test.

**Table 3 cancers-16-01259-t003:** List of thirty metabolites with absolute coefficient value > 0.1 on dimension one of the optimal 2-dimension projection of data using Penalized Discriminant Analysis (PDA).

#	Variable ID	Variable Name	Coefficient on PDA Dimension One
1	1	C2	0.15
2	4	C5.1	−0.15
3	6	C4.OH	0.29
4	7	C6	−0.13
5	8	C5.OH.C3.DC	0.14
6	9	C4.DC.Ci4.DC	−0.16
7	10	C8.1	0.13
8	11	C8	0.17
9	16	C10.2	−0.15
10	18	C10	0.2
11	19	C7.DC	−0.24
12	24	C12.OH.C10.DC	−0.11
13	25	C14.2	0.1
14	26	C14.1	−0.11
15	30	C16.2	0.17
16	34	C16.OH.C14.DC	−0.1
17	35	C18.2	−0.12
18	36	C18.1	0.33
19	37	C18	0.12
20	40	C18.OH.C16.DC	−0.22
21	41	C20.4	0.12
22	43	C18.1.DC	0.12
23	45	Cer.d18.1.14.0.	0.1
24	57	GlcCer.d18.1.16.0.	−0.11
25	58	GlcCer.d18.1.18.0.	−0.15
26	59	GlcCer.d18.1.20.0.	−0.17
27	60	GlcCer.d18.1.22.0.	−0.17
28	81	SM.d39.3.	−0.13
29	82	SM.d39.2.	−0.11
30	92	SM.d43.3.	−0.15

Note: grey highlight identifies two metabolites that significantly differed by race based on raw *p*-values < 0.05 from Wilcoxon rank-sum test.

## Data Availability

De-identified metabolomics data that support the findings of this study are available from the corresponding author upon reasonable request, as from the time of publication, and after approval of a proposal with a signed data access agreement. However, metabolomics data will eventually be deposited at www.metabolomicsworkbench.org (accessed on 19 March 2024).

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
