# Peer review of "Racial Differences in Vaginal Fluid Metabolites and Association with Systemic Inflammation Markers among Ovarian Cancer Patients: A Pilot Study"

_cancers, 2024, doi:10.3390/cancers16071259_

Round 1

Reviewer 1 Report

Comments and Suggestions for Authors The study addressed microbial metabolites with differential abundance between different racial groups and in this regard the authors have extended their previous observations. Bacterial metabolite production is a key step in regulating neoplasia progression. Racial differences are important in understanding individual differences and racial differences in risk factors.

Upon reviewing the manuscript I have not come across any major issue that requires attention.

Author Response

Thanks for reviewing our manuscript.

Reviewer 2 Report

Comments and Suggestions for Authors

It was a good thought to relate the metabolite composition to microbial diversity in the black’s vs white ovarian cancer patients vaginal fluid extracts.

But the study needs bigger datasets to draw the conclusions as most of the data was not significant. Some of the metabolites which are slightly higher they are broadly connected to the cytokines without any clear pathway or mechanism

Data representation need to be improved and the legends are missing.

Data representation need to be improved and the legends are missing.

Comments on the Quality of English Language

It was difficult to understand and follow through the story because of too many technical terms in the explanations. Need to improve the flow of story

Author Response

Comments and Suggestions for Authors

It was a good thought to relate the metabolite composition to microbial diversity in the black’s vs white ovarian cancer patients vaginal fluid extracts.

Authors’ response: Thank you.

But the study needs bigger datasets to draw the conclusions as most of the data was not significant. Some of the metabolites which are slightly higher they are broadly connected to the cytokines without any clear pathway or mechanism

Authors’ response: We agree with the observation that larger datasets are required to reach definitive conclusions. However, our study is a preliminary investigation that can inform future larger studies. Based on the results from this investigation, we speculate and make heuristic guesses on what pathways might be involved, providing justification for future investigation that would generate unequivocal evidence on biological mechanisms and pathways implicated by racial/ethnic differences in vaginal fluid metabolites among patients with ovarian cancer.

Data representation need to be improved and the legends are missing.

Data representation need to be improved and the legends are missing.

Authors’ response: Thanks for pointing this out.  We have ensured that legends are now included beneath each figure in the revised manuscript in lines 277-280 for Figure 1, 281-284 for Figure 2, 285-291 for Figure 3.

Comments on the Quality of English Language

It was difficult to understand and follow through the story because of too many technical terms in the explanations. Need to improve the flow of story

Authors’ response: Thanks for feedback. We have improved the flow of our manuscript by simplifying terms and defining required technical jargons at first use. For example, we defined ‘metabolomics’ in lines 104-106 in the introduction. We also formatted the structure and reorganized sections to improve flow. Some technical terms are mandatory (e.g., reagents used for processing samples, names of kits for sample collection). However, the purpose of using such technical terms is to ensure accurate documentation of the scientific method that would enable and ensure reproducibility and replicability.

Reviewer 3 Report

Comments and Suggestions for Authors

The authors report on the examination of racial differences in vaginal fluid metabolites and association with systemic inflammation markers among ovarian cancer patients.

They found that vaginal fluid metabolites concentrations likely differ by race, with Black individuals having a unique pattern of some down regulated metabolites including arachidonoylcarnitine, which has a known indirect role in carcinogenesis.

Comments

1.The authors cited two previous reports which mentioned that: “Ovarian cancer is the most lethal female cancer with disproportionately worse survival for Black patients despite lower incidence.2,3”.(lines 42-43).

This means that specific vaginal microbiota metabolic products, in black women, including significant higher level of arachidonoylcarnitine, are protective for the occurrence of ovarian cancer, but if, unfortunately, this disease occur, the inflammatory products aggravate its evolution. The authors should explain this aspect.

2.It is no doubt that vaginal flora can influence, or even can be responsible for pelvic inflammatory disease, endometriosis and cervical cancer, as the authors stated. (lines 53-54)

But, the authors concluded that: “Moreover, significant correlations between systemic inflammation biomarkers and targeted metabolites suggest involvement of vaginal fluid in sustained inflammation, although no racial differences were observed. We speculate that racial differences in vaginal microbiome metabolites and correlations with systemic inflammation markers may be biologically relevant to ovarian cancer through pathways involving mitochondrial dysfunction and sphingolipid metabolism.”.

Or, any study should have a clinical impact and/or a prospective value. Reading the present manuscript, the reader conclude that it is possible to improve the ovarian cancer evolution by correcting the vaginal flora composition.It is a very interesting conclusion.

Author Response

Reviewer #3

The authors report on the examination of racial differences in vaginal fluid metabolites and association with systemic inflammation markers among ovarian cancer patients.

They found that vaginal fluid metabolites concentrations likely differ by race, with Black individuals having a unique pattern of some down regulated metabolites including arachidonoylcarnitine, which has a known indirect role in carcinogenesis.

Comments

1.The authors cited two previous reports which mentioned that: “Ovarian cancer is the most lethal female cancer with disproportionately worse survival for Black patients despite lower incidence.2,3”.(lines 42-43).

This means that specific vaginal microbiota metabolic products, in black women, including significant higher level of arachidonoylcarnitine, are protective for the occurrence of ovarian cancer, but if, unfortunately, this disease occur, the inflammatory products aggravate its evolution. The authors should explain this aspect.

Authors’ response: Thanks for this comment. We appreciate the reviewers’ speculation about the big picture interpretation of observed patterns in our study and connection to cancer outcomes. That is the goal of studies like ours, to shed insight on observed differential survival by race and identify biological mechanisms that can be targeted with therapeutic advances. However, because this is a pilot study with small sample sizes, the evidence is insufficient to make unequivocal interpretations of observed patterns and what the biological implications on outcomes are. At best what we can say is that in our study sample, Black women appeared to have lower median concentrations of arachidonoylcarnitine, while White women appeared to have higher median concentrations. Based on prior studies on colon cancer (detailed in the discussion section in lines 355-366), arachidonoylcarnitine might be important in cancer progression. However, we cannot make definitive statements on what its specific role in ovarian cancer is. Instead, we provide justification for future larger studies to establish specific patterns detailing how vaginal fluid metabolites differ by race, and relating these differences to ovarian cancer survival, and exploring specific biological pathways through which metabolites may influence ovarian cancer outcomes.

2.It is no doubt that vaginal flora can influence, or even can be responsible for pelvic inflammatory disease, endometriosis and cervical cancer, as the authors stated. (lines 53-54)

But, the authors concluded that: “Moreover, significant correlations between systemic inflammation biomarkers and targeted metabolites suggest involvement of vaginal fluid in sustained inflammation, although no racial differences were observed. We speculate that racial differences in vaginal microbiome metabolites and correlations with systemic inflammation markers may be biologically relevant to ovarian cancer through pathways involving mitochondrial dysfunction and sphingolipid metabolism.”.

Or, any study should have a clinical impact and/or a prospective value. Reading the present manuscript, the reader conclude that it is possible to improve the ovarian cancer evolution by correcting the vaginal flora composition. It is a very interesting conclusion.

Authors’ response: Thanks for seeing the prospective value of our scientific contribution. We agree that our findings have potential actionable value, including the identification of prognostic biomarkers, and informing the design of population-specific therapeutic interventions. However, given that these findings are currently generated from a small-sized pilot study, a pre-requisite step for this potential value to materialize is for larger sample sized studies to be conducted. Such studies will generate unequivocal detailed evidence that will guide design of targeted therapeutic action.  Generating such evidence with larger sized studies is the focus of our research group, as we are in the process of securing funds that will enable much larger sample sizes to further investigate the role of vaginal fluid metabolites in racial disparities among ovarian cancer patients.

Reviewer 4 Report

Comments and Suggestions for Authors

This is a quite interesting research article with quite novelty. However, several points should be addressed.

- The abstract should be re-written by following a better organized structure. The authors should organized the abstract by using distinct sections, e.g. background (in which the aim of the study should be included), methods, results and conclusions.

- The second sentence of the abstract is very confusing and needs re-phrasing.

- The sample size is quite small. Thus, the authors should include in the title that this is a pilot or a preliminary study, e.g. "Examination of racial differences in vaginal fluid metabolites and association with systemic inflammation markers among ovarian cancer patients: a pilot study".

- The conclusions of the abstract are confusing and should be re-written highlighiting what future studies could be performed based on the results of the present study.

- The first paragraph of the Introduction section is too small and it needs to be enriched by more epidemiological data concerning ovarian cancer, as well as by adding some molecular pathophysiological mechanism of the disease related with inflammation and other specific pathological mechanisms governing ovarian cancer.

- The authors should include a paragraph with the most basic principles of metabolomics concept. This will be very useful for the readers. In this paragraph the authors should include some indicative application of metabolomics in cancer disease, as well as in ovarian cancer if exist.

- In the Methods section, subheadings should be added, e.g. Study design should be reported as a subheading in a separate line over the following text.

- The paragraph concerning Study design is too long and it should be split into two paragraphs.

- In the Statistical analysis section, the authors should report what normality tes they used to assess the distribution of their variables which will then support what statistical test used for their analysis.

- In general, statistical analysis section is written with a very complex manner which is not easily understood by the readers. The authors need to re-organized this section in order to be more simple in order to be nore understood by the readers.

- The authors should more clear report the ethical approval code by the Ethical Committee of their institution.

- The authors shoul explain in the statistics section the reaseon for which the select Fisher exact test and not Chi-square test for the analysis of their variables.

- Is a multivariate analysis nesessary in this type ofresearch study?

- The Table 1 should be included into the main text and not in the supplementary material. This will be very useful for the readers.

- The Table 2 is not reported that is in the supplemmentary material.

- The Figure 2 should be included into the main text and not in the supplementary material. This will be very useful for the readers.

- In general, the text of the results is quite confusing. The authors should try to present teir results with a more simple way.

- The authors should better re-organized the discussion section in order to emphasize the comparison of their results with previous studies, highlighiting also the literature gap that they covered by their results.

- Before the paragaph with the limitations of the study, the authors should add a paragraph with the strengths of their study.

- The conclusions should be written in a separate section.

- In the conclusion section, the authors should highlight what specific research could be done on the field of their studies using as a basis their results.

-References style doesn not follow thw recommended refernces style of the journal.

- The authors should try to add scertain more recent bibliography from the last 2-3 years.

- There are several English language errors as well syntax/grammar errors that make several sentences very complex. So, extensive English language editing is strongly recommended.

Comments on the Quality of English Language

Extensive editing of English language required

Author Response

Comments and Suggestions for Authors

This is a quite interesting research article with quite novelty. However, several points should be addressed.

Authors’ response: Thanks for positive comment and providing constructive feedback.

- The abstract should be re-written by following a better organized structure. The authors should organized the abstract by using distinct sections, e.g. background (in which the aim of the study should be included), methods, results and conclusions.

Authors’ response: Thanks. Abstract now has a structured format, without the headings as required by Cancers formatting instructions, in the revised manuscript (see lines 33-51 in revised manuscript).

- The second sentence of the abstract is very confusing and needs re-phrasing.

Authors’ response: Thanks for pointing this out. We rephrased both the second sentence, and the preceding sentence in the abstract (see lines 33-37) to give better context and enhance clarity. The updated sentences now read as: “The vaginal microbiome differs by race and contributes to inflammation by directly producing or consuming metabolites or by indirectly inducing host immune response, but its potential contributions to ovarian cancer (OC) disparities remain unclear. In this exploratory cross-sectional study, we examine whether vaginal fluid metabolites differ by race among patients with OC, if they are associated with systemic inflammation, and if such associations differ by race.”

- The sample size is quite small. Thus, the authors should include in the title that this is a pilot or a preliminary study, e.g. "Examination of racial differences in vaginal fluid metabolites and association with systemic inflammation markers among ovarian cancer patients: a pilot study".

Authors’ response: Thanks for pointing this out. We have now modified the title (lines 1-4) to read as suggested: "Racial differences in vaginal fluid metabolites and association with systemic inflammation markers among ovarian cancer patients: a pilot study".

- The conclusions of the abstract are confusing and should be re-written highlighiting what future studies could be performed based on the results of the present study.

Authors’ response: Thanks. Conclusion in abstract (lines 48-52) has been modified to reflect this comment; it now reads as: “These findings suggest that vaginal fluid metabolites may differ by race, are linked with systemic inflammation, and hint at a potential role for mitochondrial dysfunction and sphingolipid metabolism in OC disparities.  Larger studies are needed to verify these findings, and further establish specific biological mechanisms that may link the vaginal microbiome with OC racial disparities”

- The first paragraph of the Introduction section is too small and it needs to be enriched by more epidemiological data concerning ovarian cancer, as well as by adding some molecular pathophysiological mechanism of the disease related with inflammation and other specific pathological mechanisms governing ovarian cancer.

Authors’ response: Thanks. We have extended paragraph one in the introduction to include several epidemiological facts on ovarian cancer in lines 57-65.

Also, we describe how vaginal microbiome-induced inflammation may contribute to tumor growth by adding another paragraph to lines 80-93 in the introduction.

- The authors should include a paragraph with the most basic principles of metabolomics concept. This will be very useful for the readers. In this paragraph the authors should include some indicative application of metabolomics in cancer disease, as well as in ovarian cancer if exist.

Authors’ response: This suggestion has been incorporated into the introduction section by including a paragraph in lines 104-116 focused on metabolomics approaches and its application to cancer investigations.

- In the Methods section, subheadings should be added, e.g. Study design should be reported as a subheading in a separate line over the following text.

Authors’ response: Thanks. The following subheadings have been added to the Methods section (lines 121-232): 2.1 Study design, 2.2 Biospecimen collection and processing, 2.3 Analytical methods with subheadings for Acylcarnitines, Ceramides and Sphingomyelins, conventional metabolites, and assays for biomarkers of systemic inflammation, and 2.4 Statistics.

- The paragraph concerning Study design is too long and it should be split into two paragraphs.

Authors’ response: Thanks. The previous study design subsection has been split into two sections in this revised manuscript, consisting of one subsection explicitly focused on Study design (lines 122-132), and one subsection focused on biospecimen collection and processing (lines 133-146).

- In the Statistical analysis section, the authors should report what normality tes they used to assess the distribution of their variables which will then support what statistical test used for their analysis.

Authors’ response: Thanks. Our study sample size is small (N=36). Hence, there was no compelling reason to test for normality, given that assumptions of normality cannot be adequately assessed for small sample sizes.(Altman, Gore, Gardner, & Pocock, 1983) While we transformed our metabolite data by log-transforming and auto-scaling, we opted to use a non-parametric statistical test that is “invariant to transformations and shows excellent operating characteristics in situations where parametric tests have optimal power” (Frank Harrell, 2014).  We have reflected this reasoning in the manuscript text in lines 194-199.

- In general, statistical analysis section is written with a very complex manner which is not easily understood by the readers. The authors need to re-organized this section in order to be more simple in order to be nore understood by the readers.

Authors’ response: We have simplified the statistical analyses section in lines 193-232 by i. reorganizing to consist of the following subsections in the following order: descriptive summary, data-processing steps, racial differences in vaginal fluid metabolites, correlations with biomarkers of systemic inflammation, and software, and ii. clearly defining technical jargons that are referred to and giving reasons for using specific statistical tests.

- The authors should more clear report the ethical approval code by the Ethical Committee of their institution.

Authors’ response: Thanks. Ethical approval code by the Institutional Review Board of Duke University Health System is now reported in the manuscript in lines 493-496.

- The authors shoul explain in the statistics section the reaseon for which the select Fisher exact test and not Chi-square test for the analysis of their variables.

Authors’ response: Thanks. We appropriately used Fisher’s exact test instead of Chi-square test due to small sample sizes, since Chi-square approximation is inadequate when cells with values < 5 are present, which is the situation in our small sample sized dataset. Therefore, we have now added this sentence to lines 196-197 in order to indicate why Fisher’s exact test was used: “Fisher’s exact test effectively handles low expected cell counts less than 5 typical of small sized datasets”.

- Is a multivariate analysis nesessary in this type of research study?

Authors’ response: Yes. In the context of our research study, while univariate analysis shows proof of how each metabolite might differ by race, a multivariate approach is needed to simultaneously consider all metabolites together and get a full, more representative, picture of overall differences in targeted metabolites considered (Worley & Powers, 2013). We can see this with the Penalized Discriminant Analyses, which shows overall how Black and White patients cluster together using information from all 99 metabolites. Moreover, multivariate methods are necessary to accommodate the large K small n nature of metabolomics dataset, which implies an excess of observed variables (i.e., K) relative to number of observations (i.e., n) (Worley & Powers, 2013).

Authors’ response:

- The Table 1 should be included into the main text and not in the supplementary material. This will be very useful for the readers.

Authors’ response: Thanks. eTable 1 has been moved to the main text and is now Table 3, referred to in line 276.

- The Table 2 is not reported that is in the supplemmentary material.

Authors’ response: Thanks for picking this up. This table has been renamed as eTable 1, and the revised manuscript now includes a reference to eTable 1 on lines 304, and 306.

- The Figure 2 should be included into the main text and not in the supplementary material. This will be very useful for the readers.

Authors’ response: Thanks. We have moved Supplementary Figure S2 to the main text, and is now Figure 2 along with its legend in the main text on lines 281 - 284.

- In general, the text of the results is quite confusing. The authors should try to present teir results with a more simple way.

Authors’ response: We have simplified our results presentation, in lines 233-315, by organizing into sections: i. descriptive summary, ii. racial differences, and iii. correlations with biomarkers of inflammation.

- The authors should better re-organized the discussion section in order to emphasize the comparison of their results with previous studies, highlighiting also the literature gap that they covered by their results.

Authors’ response: Thanks. This suggestion has been incorporated by focusing the second paragraph of the discussion on lines 333-354 to describe prior reports, and relate our findings to the gap in the literature.

- Before the paragaph with the limitations of the study, the authors should add a paragraph with the strengths of their study.

Authors’ response: Thanks. This comment has been addressed by stating the strengths of our study at the start of the paragraph on the limitations of the study on lines 400 – 404.  These lines read as follows:

“A major strength of our study is its novelty, as this is the first study to our knowledge that has investigated racial disparities by examining racial differences in vaginal fluid metabolites. Our study provides valuable pilot data for informing future studies and demonstrates the viability of examining the vaginal microbiome’s functional role in ovarian cancer health disparities.”

- The conclusions should be written in a separate section.

Authors’ response: Thanks. We have moved the conclusion to a separate section in lines 428-440.

- In the conclusion section, the authors should highlight what specific research could be done on the field of their studies using as a basis their results.

Authors’ response: Thanks. We modified the last two concluding sentences in lines 437-440 to reflect this comment: “Future studies should evaluate whether these findings are reproducible in a larger sample, and establish what their specific biological implications are. This study pro-vides a template for further investigation of how inflammation from vaginal fluid metabolites may be related to disparities in ovarian cancer.”

-References style doesn not follow thw recommended refernces style of the journal.

Authors’ response: Thanks. We have modified the reference style to the numbered style throughout the manuscript text, as required by Cancers.

- The authors should try to add scertain more recent bibliography from the last 2-3 years.

Authors’ response: Thanks. We have added the following six relevant references that were published within the last 2-3 years:

Asangba, A.E., et al., Diagnostic and prognostic potential of the microbiome in ovarian cancer treatment response. Scientific Reports, 2023. 13(1): p. 730.

Zhao X, Liu Z, Chen T. Potential role of vaginal microbiota in ovarian cancer carcinogenesis, progression and treatment. Pharmaceutics. 2023;15(3):948.

Dhingra, A., et al., Microbiome and Development of Ovarian Cancer. Endocrine, Metabolic & Immune Disorders-Drug Targets (Formerly Current Drug Targets-Immune, Endocrine & Metabolic Disorders), 2022. 22(11): p. 1073-1090.

Peres, L.C. and J.M. Schildkraut, Racial/ethnic disparities in ovarian cancer research. Advances in cancer research, 2020. 146: p. 1-21.

Anyanwu, M.C., et al., Race, Affordability and Utilization of Supportive Care in Ovarian Cancer Patients. Journal of Pain and Symptom Management, 2022.

Montes de Oca, M.K., et al., Healthcare Access Dimensions and Guideline-Concordant Ovarian Cancer Treatment: SEER-Medicare Analysis of the ORCHiD Study. J Natl Compr Canc Netw, 2022. 20(11): p. 1255-1266.e11.

We will gladly include any further specific recommendations available from the reviewer. Also, of the 60 cited references in our bibliography, 29 (~49%) were authored between 2020 and 2023. This pattern reflects the need for more in-depth investigation of the on the subject, which is what our study aims to achieve.

- There are several English language errors as well syntax/grammar errors that make several sentences very complex. So, extensive English language editing is strongly recommended.

Authors’ response: We apologize for this, and thank you for your observations. This comment has now been addressed by incorporating reviewers’ suggestions, and by engaging a native English speaker, Lauren Wilson, PhD to read and improve the grammar in our reporting.

Comments on the Quality of English Language

Extensive editing of English language required

Authors’ response: Thanks. This comment has been addressed as described in the comment immediately above.

References

Altman, D. G., Gore, S. M., Gardner, M. J., & Pocock, S. J. (1983). Statistical guidelines for contributors to medical journals. British medical journal (Clinical research ed.), 286(6376), 1489.

Worley, B., & Powers, R. (2013). Multivariate Analysis in Metabolomics. Curr Metabolomics, 1(1), 92-107. doi:10.2174/2213235x11301010092

Round 2

Reviewer 2 Report

Comments and Suggestions for Authors

Most of the data is based on the predictions, very limited data supports the assumptions. Only a single cohort of metabolites are differing and there is no mechanistic evidence or any additional experiments supporting the same.

More experimental evidence are required at this stage to draw conclusions, which would provide sufficient evidence for the future research

Reviewer 4 Report

Comments and Suggestions for Authors

The authors have significantly improved their manuscript. I would like to thank them for their trust.

Comments on the Quality of English Language

Minor editing of English language required